# A Cinnamate 4-HYDROXYLASE1 from Safflower Promotes Flavonoids Accumulation and Stimulates Antioxidant Defense System in Arabidopsis

**DOI:** 10.3390/ijms24065393

**Published:** 2023-03-11

**Authors:** Yuying Hou, Yufei Wang, Xiaoyu Liu, Naveed Ahmad, Nan Wang, Libo Jin, Na Yao, Xiuming Liu

**Affiliations:** 1College of Life Sciences, Engineering Research Center of the Chinese Ministry of Education for Bioreactor and Pharmaceutical Development, Jilin Agricultural University, Changchun 130118, China; 2Institute of Life Sciences, Wenzhou University, Wenzhou 325035, China; 3Biomedical Collaborative Innovation Center of Zhejiang Province, Wenzhou University, Wenzhou 325035, China

**Keywords:** cinnamate 4-hydroxylase, safflower, flavonoid biosynthesis, antioxidant, Arabidopsis

## Abstract

C4H (cinnamate 4-hydroxylase) is a pivotal gene in the phenylpropanoid pathway, which is involved in the regulation of flavonoids and lignin biosynthesis of plants. However, the molecular mechanism of C4H-induced antioxidant activity in safflower still remains to be elucidated. In this study, a CtC4H1 gene was identified from safflower with combined analysis of transcriptome and functional characterization, regulating flavonoid biosynthesis and antioxidant defense system under drought stress in Arabidopsis. The expression level of CtC4H1 was shown to be differentially regulated in response to abiotic stresses; however, a significant increase was observed under drought exposure. The interaction between CtC4H1 and CtPAL1 was detected using a yeast two-hybrid assay and then verified using a bimolecular fluorescence complementation (BiFC) analysis. Phenotypic and statistical analysis of CtC4H1 overexpressed Arabidopsis demonstrated slightly wider leaves, long and early stem development as well as an increased level of total metabolite and anthocyanin contents. These findings imply that CtC4H1 may regulate plant development and defense systems in transgenic plants via specialized metabolism. Furthermore, transgenic Arabidopsis lines overexpressing CtC4H1 exhibited increased antioxidant activity as confirmed using a visible phenotype and different physiological indicators. In addition, the low accumulation of reactive oxygen species (ROS) in transgenic Arabidopsis exposed to drought conditions has confirmed the reduction of oxidative damage by stimulating the antioxidant defensive system, resulting in osmotic balance. Together, these findings have provided crucial insights into the functional role of CtC4H1 in regulating flavonoid biosynthesis and antioxidant defense system in safflower.

## 1. Introduction

Safflower (*Carthamus tinctorius* L.) is used in Chinese traditional medicine to improve and treat various diseases, such as gynecology, cerebrovascular and cardiovascular diseases, hypertension, and coronary heart disease [1]. The main component of flavonoids known as hydroxy safflower yellow A (HSYA), is a C-glucosyl quinochalcone which possess several pharmacological effects [2]. Other classes such as anthocyanins, proanthocyanidins (PAs), and flavonols are the main types of flavonoids found in almost all higher plants and are essential for plant growth development, metabolism and defense system [3]. Anthocyanins are naturally occurring water-soluble flavonoid pigments that play an important role in color formation in flowers, stems, leaves, and fruits [4]. Several studies have also demonstrated the role of anthocyanins in promoting high antioxidant capacity that can protect the plant system from free radical damage [5]. However, the explicit molecular mechanism of drought stress alleviation by exploiting the specialized metabolic system and antioxidant defense system in safflower is still not clear.

The phenylpropanoid metabolic pathway is essential to produce phenolic compounds, flavonoids, lignans, and other secondary metabolites [6]. Most of these synthesized secondary metabolites act as antioxidants and antibiotic substances that help plants resist drought stress and reduce damage caused by oxidative stress [7]. Anthocyanins and other flavonoids are directly derived from phenylalanine. The conversion process of phenylalanine to coumarin coenzyme A is catalyzed by phenylalanine ammonia lyase (PAL), cinnamate-4-hydroxylase (C4H), and 4-cinnamate coenzyme A ligase (4CL) during the phenylpropanoid pathway [8]. Cinnamic acid 4-hydroxylase (C4H) is a cytochrome 450 monooxygenase that produces p-coumaric acid at the C4 position by hydroxylating the aromatic ring of t-cinnamic acid. Previous studies have suggested that plant seeds with the C4H mutation lead to a significant decrease in anthocyanins, growth hormones, and lignin in Arabidopsis [9]. Similarly, the abundance of C4H expression at the transcriptional level and protein level can significantly influence the biosynthesis of aromatic compounds and flavonoids in plants [10,11]. These findings have provided valuable insights towards the importance of C4H in plants that can affect the synthesis and accumulation of flavonoids and phenolic compounds by regulating its expression level along their biosynthetic pathway.

Drought is considered as one of the most important factors affecting plant growth and yield, having a major impact on gene expression, morphology, metabolic adjustments, and photosynthetic modifications. Several studies have demonstrated that plant growth and development can be hampered by osmotic and oxidative stress, ion deficiencies, and metabolic disturbances when water is scarce [11]. Similarly, severe water deficiency has a significant negative impact on the growth and development of safflower [12]. A decrease in water availability during nutritional growth has a significant impact on chlorophyll content, film stability, and leaf area, all of which can affect the overall yield. Importantly, drought stress harms physiological status such as leaf temperature, osmotic adjustment, and stomatal conductance in safflower. Drought stress also increases the accumulation of total soluble sugars, proline, ascorbic acid, anthocyanins, and secondary metabolites, all of which act as antioxidants to reduce oxidative damage in plants by detoxifying excess ROS [13]. ROS is an inevitable by-product of aerobic metabolism and a key signal molecule that enables cells to respond quickly to different stimuli. In plants, ROS can sense abiotic and biotic stresses and activate stress response networks, thus contributing to the establishment of defense mechanisms and plant resilience, and plays an important role in regulating growth metabolism and environmental challenges [14,15]. The production of ROS during water stress can lead to an imbalance between oxidative free radicals and antioxidant mechanisms. Studies have shown that ROS in excess may cause abnormal cell signals, membrane damage, and cell death in plants. By reducing oxidative free radicals, the degree of membrane damage, cell acidity and toxicity can be controlled and cell metabolism can be stabilized through the preservation of macromolecule integrity [16,17]. Therefore, it is of great practical significance to investigate the mechanism and strategic role of C4H in antioxidant defense system in safflower.

In the present work, a candidate CtC4H1 gene was identified from safflower using transcriptome data and several functional approaches, which was responsive to various stresses, such as drought, MeJA, and lignt stress. The characterization of CtC4H1 was carried out using molecular cloning, phylogenetic analysis, subcellular localization, expression profiling under normal and drought stress conditions, overexpression in *Arabidopsis thaliana*, phenotyping and physiological data analysis of the transgenic plants. In addition, we also investigated the level of antioxidant enzyme system and expression level of key structural genes of flavonoid pathways in transgenic plants. Together, our findings have revealed that the overexpression of CtC4H1-encoding gene has drought tolerance by reducing oxidative damage and increasing secondary metabolism in Arabidopsis. Altogether, these findings have provided crucial insights into the functional role of CtC4H1 in regulating flavonoid biosynthesis and the antioxidant defense system in safflower.

## 2. Results

### 2.1. Transcriptome Analysis and Identification of CtC4H Genes in Safflower

A total of 128 differentially expressed genes (DEGs) were identified by analyzing the transcriptome data of different flowering stages of safflower. Of these DEGS, 91 were detected with up-regulation and 37 were down-regulated in different flowering stages of safflower. Importantly, the genes involved in flavonoid biosynthetic pathway demonstrated preferential expression level through different flowering stages. However, the expression level of genes involved in the upstream pathway of flavonoids were slightly expressed except for C4H-encoding genes. Based on the FPKM ratios, we identified two CtC4H encoding genes, of which, CtC4H1 showed high abundance at the full and initial flowering stages, whereas CtC4H2 expression was suppressed in full and fade flowering (Figure 1A). To further validate the integrity and reliability of the transcriptome data, the expression level of the two C4H encoding genes was investigated using qRT-PCR analysis. The results indicated a consistent pattern with the transcriptome data, demonstrating an increased expression level of CtC4H1 at the full stage followed by initial flowering. On the contrary, CtC4H2 showed a lower expression during the bud and initial flowering, while showing a down-regulated expression at the full and fading stages (Figure 1B), CtC4H1 was cloned (Appendix A) for further research.

### 2.2. Functional Annotations of DEGS

We performed GO enrichment analysis of the identified DEGs from transcriptome data, which classified them into three functional categories including biological process (BP), molecular function (MF), and cellular component (CC) (Figure 2A). The GO enrichment analysis was calculated considering the *p*-value <= 0.05 for significance. In the biological process category, the most significant GO-enriched terms included cellular process, metabolic process, responses to stimuli and biological regulation. Similarly, the molecular function category showed the enrichment of the GO terms such as catalytic activity, binding, transporter activity and transcription regulator activity. In the case of cellular component, the GO terms of cell, membrane, membrane part and organelle were significantly enriched (Appendix A). Furthermore, the top 20 functional KEGG enriched pathways were also screened out. The results suggested that the highest number of DEGs were significantly enriched into the functional pathways including plant hormone signal transduction, flavonoid biosynthesis, diterpenoid biosynthesis, peroxisome and isoflavonoid biosynthesis (Figure 2B). The functional annotation analysis of DEGs suggested valuable hallmarks towards their important functional roles in regulating crucial biological processes in safflower.

### 2.3. Phylogenetic Analysis and Multiple Sequence Alignments of CtC4H Proteins

To investigate the evolutionary relationship of CtC4H proteins with other C4H members from different plant species (Appendix A), a phylogenetic tree was constructed. The results demonstrated that all members of C4H proteins from different plant species along with CtC4H can be grouped into three main clades. Noticeably, the members of CtC4H proteins were clustered together with other members of C4H proteins from *Helianthus annuus*, *Cynara cardunculus VaR*, *Camellia sinensis*, and *Cirsium japonica*, suggesting that CtC4H-encoding genes may evolved from these terrestrial plants during the course of evolution (Figure 3A). Furthermore, the amino acid sequence of the CtC4H1 gene (Accession: OP207882) and *Chrysanthemum* (CbC4H), *Camellia sinensis* (CsC4H), *Arabidopsis thaliana* (AtC4H), *Arctium lappa* (AlC4H), *Helianthus annuus* (HaC4H), *Nicotiana tabacum* (NtC4H), and *Eupatorium adenophorum* (AaC4H) were compared using DNAMAN 9.0 software. The multiple sequence alignment and homology analysis of the C4Hs proteins indicated approximately 90% similarities with each other, suggesting that the C4H may retain strong conservation during process of evolution (Figure 3B). In addition, these members of C4H proteins shared the conservative of a cytochrome P450 domain, anchor region, proline-rich regions, threonine-containing binding, Heme binding region, and an E-R-R triad domain. Noticeably, CtC4H1 contains six unique substrate recognition sites including (SRS1, SRS2, SRS3, SRS4, SRS5, and SRS6), which showed maximum proximity with the CYP73A subfamily.

### 2.4. CtC4H1 Interacts with CtPAL1 In Vivo

The regulation of flavonoid biosynthesis is likely involved in active interaction between the major enzymes implicated in this pathway [18]. Therefore, to determine the interaction partner proteins of CtC4H1, we predicted the protein interaction network of CtC4H1 using the online webserver of STRING software (Appendix A). The results suggested that CtC4H1 directly interacts with CtPAL1 (phenylalanine ammonia lyase1) catalyzes the first step of phenylpropanoid pathway converting phenylalanine to cinnamic acid and then cinnamate-4-hydroxylase catalyzes it to coumarin [19]. To further validate their interaction, yeast-2 hybrid assay was carried out using pGBKT7-CtC4H1 and pGADT7-CtPAL1 using X-α-gal-containing selective media. All three co-transformation experimental groups included positive control: pGBKT7-p53+ pGADT7-T, negative control pGBKT7-lam+ pGADT7-T, and experimental group pGBKT7-CtC4H1+ pGADT7-CtPAL1 were allowed to grow on SD/-Leu/-Trp medium and then SD/-Leu/-Trp/-His/-Ade+X-α-gal medium. The positive control changed color to blue whereas the colonies of pGBKT7-CtC4H1+ pGADT7-CtPAL1 construct grew faster and exhibited a more obvious blue color, implying a potential interaction between the C4H1 protein and the CtPAL1 protein (Figure 4A).

To further verify the interaction between the two groups of proteins and their sites of action, a BiFC assay was performed to determine whether CtC4H1 could bind to CtPAL1, and form protein complexes in vivo. The interaction signal of pxy106-CtC4H1+ pxy104-CtPAL1 was detected using laser confocal scanning microscopy in both the nucleus and the plasma membrane, which could be attributed to the transmembrane structure. However, no signals were observed in the control pxy106-CtC4H1+ pxy104 and pxy104-CtPAL1+ pxy106 combinations (Figure 4B). Thus, the results of yeast two-hybrid assays and BiFC analysis demonstrated strong evidence on the interaction between CtC4H1 and CtPAL1 in vivo. It is expected that the co-localization of CtPAL1 and CtC4H1 in tobacco forms a metabolic channel for cinnamic acid [20], which may account for enhanced cinnamate biosynthesis in safflower. However, the specific molecular mechanism needs to be validated further.

### 2.5. Subcellular Localization Assays

The subcellular localization CtPAL1 and CtC4H1 was investigated using transient expression system each with their corresponding plant overexpression vector (pCAMBIA1300-35S-GFP) in tobacco leaves. The green, fluorescent signals of the pCAMBIA1300-CtC4H1-GFP and pCAMBIA1300-CtPAL1-GFP were dispersed in the nucleus and plasma membrane (Figure 5). These findings implied that CtPAL1 and CtC4H1 expressed simultaneously in the nucleus or plasma membrane and could regulate their corresponding substrates during phenylpropanoid pathway in safflower.

### 2.6. Expression Pattern of CtC4H1 Gene under Different Abiotic Stresses in Safflower

We investigated the expression level of CtC4H1 in safflower under different abiotic stress conditions such as drought, MeJA, light, and dark stress at different treatment times using qRT-PCR analysis. As shown in Figure 6, no significant changes were observed in the expression level of the CtC4H1 gene in the first 48 h of external drought stress; however, the expression of CtC4H1 was significantly up-regulated in a continuous manner after 60 h of drought treatment when compared to the control treatment (0 h). These findings suggested that CtC4H1 may regulate the response mechanism of safflower against drought stress. In addition, the CtC4H1 expression under MeJA treatment was steadily increased and reached its maximum at 4 h after treatment, then declined at 6 h and 8 h after treatment. Furthermore, the expression level of CtC4H1 induced by light treatment demonstrated a periodic expression pattern, suggesting up-regulation at 24 h, 48 h, and 60 h, whereas down-regulation occurred at 12 h and 13 h time periods. In contrast, the expression of CtC4H1 decreased under dark treatment showing an inhibition at almost all-time intervals. These findings led us to conclude that selective regulation of CtC4H1 expression occurs under different stresses; nonetheless, the steady increase in its expression level under drought stress provides a valuable insight into the antioxidant defense regulatory mechanism of safflower.

### 2.7. Phenotypic Analysis and Quantification of Metabolites in CtC4H1 Overexpressed Arabidopsis

To establish a stable transformation system and generation of transgenic plants, we carried out the genetic transformation of CtC4H1 in the model Arabidopsis thaliana using the floral dip transformation method. Transgenic lines were initially selected using BASTA spray and then confirmed with PCR analysis (Appendix A). A total of 12 transgenic lines were screened, of which two stable overexpression lines were selected on the basis of a high degree of CtC4H1 expression using a qRT-PCR assay (Appendix A). The phenotyping and statistical data of transgenic plants revealed that CtC4H1 overexpressed Arabidopsis demonstrated slightly wider leaves, and longer and earlier stem development (Figure 7A–E). In addition, the total flavonoids and anthocyanin contents also demonstrated an increased level in overexpressed lines compared with WT plants. These findings suggested that CtC4H1 may play a key role in regulating plant development and modulating the defensive system by accumulating specific metabolites in transgenic Arabidopsis.

### 2.8. Overexpression of CtC4H1 and CtPAL1 Enhanced Drought Stress Tolerance in Transgenic Arabidopsis

As shown previously, the interaction between CtPAL1 and CtC4H1 was detected as both of these enzymes are located upstream in the flavonoid pathway. Therefore, we also chose CtPAL1 (previously generated in the laboratory), together with CtC4H1 overexpression lines to further study the effect of these two genes in the overexpression lines under drought stress. The phenotypic analysis of two CtC4H1 overexpression lines (OE-4 and OE-10) and CtPAL1 overexpression line (OE-6) after 14 days of drought stress demonstrated slight recovery in drought-damaged leaves as compared to WT Arabidopsis (Figure 8A). Furthermore, to confirm the antioxidant defense potential of transgenic plants overexpressed with CtPAL1 and CtC4H1, we stained the leaf discs of transgenic plants along with the WT Arabidopsis with 3,3′-diaminobenzidine (DAB) and nitro blue tetrazolium (NBT) to detect the presence of the free radicals (H_2_O_2_) and (O_2_^−^), respectively (Figure 8B,C). The results of DAB and NBT staining suggested that transgenic plants accumulated less (H_2_O_2_) and (O_2_^−^) in the stressed leaves than the control compared to WT Arabidopsis.

Flavonoids detoxify the accumulation of ROS in drought-stressed plants [13]. Hence, we also examined the content of total flavonoids and anthocyanins in transgenic lines and showed a significant increase after drought stress than the control group when compared to WT Arabidopsis (Figure 8D,E). Moreover, the MDA level was also quantified in the leaf extracts of the transgenic plants and WT Arabidopsis and the result indicated significantly less MDA content in transgenic plants under drought stress than in the control when compared to the WT (Figure 8F). This indicates the greater ROS accumulation in the WT plants compared to the OE lines. Together, these findings suggest that the CtC4H1 could play an essential role in promoting drought stress tolerance by inducing the expression of other flavonoid biosynthesis genes, resulting in an enhanced accumulation of flavonoids and stimulating the antioxidant defense system.

### 2.9. CtC4H1 Positively Regulates ROS Scavenging Ability under Drought Stress

To investigate CtC4H1-induced ROS scavenging activities in CtC4H1 and CtPAL1 overexpression lines under drought stress, we analyzed the activity of SOD, CAT and POD, which are mainly involved with ROS detoxification. The result showed that OE lines increased SOD, CAT and POD activity more under drought condition than the control when compared to WT Arabidopsis (Figure 9A–C). Similarly, when compared to WT, the accumulation of H_2_O_2_ and O_2_^−^ radicals were reduced significantly in CtC4H1 and CtPAL1 OE lines under drought stress than in the control plants. (Figure 9D,E). This trend was found to be consistent with the results of the NBT and DAB staining described previously (Figure 8B,C). In addition, we also investigated the content of proline in transgenic plants along with the WT under drought conditions because it can regulate the osmotic balance in the plant cytoplasm and also stabilize the biological macromolecular structure [20]. The results of OE lines showed significantly higher proline content under drought stress than the control when compared to WT (Figure 9F). Conclusively, these results suggest that CtC4H1 could promote the antioxidant defense system of the transgenic plants, which may lead to ROS clearance in Arabidopsis.

### 2.10. CtC4H1 Positively Regulates the Expression of Genes Involved in Flavonoid Biosynthesis

To further confirm CtC4H1-induced regulation of other key structural genes involved in the downstream pathway of flavonoid pathway, the expression analysis of nine genes was investigated in OE transgenic lines using qRT-PCR analysis. The results indicated almost all of the key genes involved in the flavonoid pathway showed up-regulation in OE-transgenic lines when compared to WT Arabidopsis (Figure 10). Noticeably, the gene expression level of AtPAL was abundantly up-regulated in both OE-transgenic lines, consistent with our previous findings of protein interaction network between CtC4H and CtPAL. In addition, the relative fold changes in transcript abundance of the downstream regulatory genes such as AtF3H, AtDFR, AtFLS, and AtANS showed an enhanced expression in OE lines when compared with WT. Thus, these findings significantly highlight the potential regulatory role of the CtC4H1 gene promoting the flavonoid biosynthetic pathway.

## 3. Discussion

Cinnamate 4-hydroxylase is the first monooxygenase (CYPP450) in the phenylpropanoid metabolic pathway; it catalyzes the hydroxylation of trans cinnamate into p-coumaric acid. The monooxygenase reaction is critical in the biosynthesis of plant phenylalanine, anthocyanins, alkaloids, terpenoids, and other metabolites. Studies have demonstrated that these metabolites are regulated by most of the members of the CYP73 family, which can mediate the synthesis of flavonoids fatty acids and lignin [21,22]. C4H is a typical P450 enzyme distributed across the plant species and belongs to the CYP73 family [23]. It is the first cytochrome P450 monooxygenase (CYP450s) to be discovered, cloned, and confirmed during flavonoid biosynthesis in plants [24]. However, the molecular mechanism of C4H-induced flavonoid biosynthesis and its role in regulating antioxidant defense responses in safflower has not yet been explored. Understanding the mechanism and strategic function of C4H in the antioxidant defense system in safflower is, therefore, of major practical importance. In this study, we have discovered a putative CtC4H1 encoding gene from safflower and demonstrated its role in drought stress tolerance via regulating flavonoid biosynthesis and the antioxidant defense system in Arabidopsis using combined transcriptome and functional analysis.

Environmental stress can disrupt the biosynthesis of secondary metabolites and reduce their diversity by affecting the expression level of their regulatory genes. In plants, elicitors are environmental cues that are communicated via plant growth regulators and other signaling chemicals such as salicylic acid (SA), jasmonic acid (JA), and gibberellin (GA) [25]. However, secondary metabolites like phenylpropanoids may also protect plants from biotic and abiotic stresses. Under environmental stress, rapid induced transcription of genes responsible for secondary metabolite production may increase the accumulation of these metabolites. Recent research has confirmed that secondary metabolite production, gene expression, and post-translational modification are the primary responses of plants to environmental stress. Similarly, external factors have been shown to influence the regulation of flavonoids biosynthesis under UVB radiation, high temperature, drought, low temperature, as well as hormonal signals such as jasmonic acid, abscisic acid, and BA [26]. In this study, we also observed the expression level of CtC4H1 in safflower under different abiotic stress conditions such as drought, MeJA, light, and dark stress at different treatment times. A significantly higher level of CtC4H1 expression was observed after 48 h of drought treatment compared to the control treatment. Light treatment triggered a periodic shift in the expression pattern of CtC4H, whereas MeJA treatment resulted in a steady increase at 4 h after treatment and a subsequent decline at 6 and 8 h after treatment. On the other hand, the CtC4H1 expression under dark treatment showed down-regulation and resulted in extreme insensitivity towards dark stress. Based to these results, we concluded that CtC4H1 expression is selectively regulated against diverse stressors; however, the consistent upregulation of CtC4H1 expression during drought stress offers a vital insight into the regulatory mechanism driving the antioxidant defense in safflower.

The accumulation of flavonoids is closely linked to stress responses, growth, and developmental signaling in several plant species. Several studies have demonstrated that lignin enhances the mechanical strength of plant cell walls and promotes growth regulation [27]. Similarly, it was discovered that the accumulation of flavonoids and the expression of C4H increased or decreased synchronously at different stages of fruit development in BlackBerry [28]. Another study revealed that C4H expression was abundant with high lignification in parsley pedicels [29]. Functional investigation of C4H showed that the mRNA accumulation level appeared to be consistent with PAL in response to stimulation throughout different phases of higher plant growth and development and/or in response to diverse external stressors such as mechanical damage, fungal infection, chemical inducers, elicitors, etc. [30]. Furthermore, defects in PAL and C4H can reduce the flux of phenyl propionic acid production, including lignin monomers and flavonoids. Arabidopsis has only one copy of the C4H gene (C4H, At2g30490). The Arabidopsis ref3 mutant is a sub morphological mutation carrying C4H that causes overall changes in phenylpropanoid biosynthesis as well as developmental abnormalities such as dwarfs. Simultaneously, inactivation of the AtC4H protein in Arabidopsis ref3 mutant reduces the content of lignin and flavonoids in plants [9,31]. In this study, we also compared the phenotypes of WT and CtC4H OE lines and the results suggested that the OE lines demonstrate slightly larger and wider leaves with longer and earlier stem development. In addition, the combined flavonoid and anthocyanin contents were likewise higher in overexpressed lines than in WT plants. These results revealed that CtC4H1 accumulates particular metabolites, suggesting a potential function in controlling plant growth and influencing the defense system in transgenic Arabidopsis.

Drought stress causes oxidative stress in plants by increasing the levels of ROS. Excessive ROS participate in a variety of signal pathways, can cause extensive cell damage and, eventually, cell death [32]. Studies have demonstrated that premature leaf senescence may result in insufficient reuse of stored metabolites from older tissues to growing tissues, affecting yield and quality of safflower [33]. As a consequence, the enhancement of the antioxidant system and the phenyl propionic acid pathway could play an essential role in delaying plant senescence [28]. Additionally, the regulation of flavonoid biosynthesis can also overcome excessive ROS accumulation under long-term drought stress conditions, which switches on the antioxidants defense system in plants [27]. Several studies confirmed that drought, light, high temperature, low-temperature pathogens, and other biotic and abiotic stresses, plus hormones, can activate the flavonoid flux to varying degrees in plants [34,35]. The plant system can eliminate drought-induced ROS by enhancing cell membrane stability and maintaining redox homeostasis, allowing plants to mitigate drought stress [36,37,38]. Previous studies also demonstrated that the activities of flavonoid metabolic enzymes (PAL, C4H) in soybean treated with UV-B radiation showed a higher flavonoid content than in the control [39,40]. In the same way, ultraviolet radiation induced the expression of PAL and C4H genes, which was consistent with an increase in the level of phenolic compounds in tomato fruit, signifying the strong interaction between genes encoding PAL and C4H [41]. In this study, we also investigated the effect of CtPAL1 and CtC4H1 in the overexpression transgenic lines under drought stress. When compared to WT, the CtC4H1 OE (OE-4 and OE-10) and a CtPAL1 OE line (OE-6) showed mild recovery in leaf shape following 14 days of drought stress. In addition, the DAB and NBT staining revealed less accumulation of free radicals (H_2_O_2_) and (O_2_^−^) in OE lines subjected to drought stress than the control compared to WT Arabidopsis. Similarly, the content of total flavonoids and anthocyanins in transgenic lines suggested significant increase under drought stress (Figure 8D,E), whereas the MDA level was found significantly less in transgenic plants under drought stress (Figure 8F). Moreover, the CtC4H1-induced ROS scavenging activities showed that OE lines increased SOD, CAT and POD activity under drought condition whereas the accumulation of H_2_O_2_ and O_2_^−^ radicals were reduced significantly. The content of proline in transgenic plants under drought condition had a significantly higher proline content under drought stress than the control when compared to WT. Altogether, our results suggested that CtC4H1 may play an important role for drought stress tolerance by inducing the expression of key flavonoid biosynthesis genes, resulting in enhanced accumulation of flavonoids and stimulating the antioxidant defense system via ROS scavenging in Arabidopsis.

Flavonoid biosynthesis, its composition, and transport mechanism in the model plant Arabidopsis still remains challenging and demands appropriate gate-ways. The multi-channel regulation of flavonoid-related core structural genes is one potential strategy. Prior studies revealed that when the expression of PAL increased, more phenylalanine molecules were converted into cinnamic acid, providing more precursor molecules for the flavonoid pathway. This may be due to the feedback effect of excess trans-cinnamic acid consumption accelerating the expression of the Arabidopsis phenylalanine ammonia lyase AtPAL, resulting in greater phenylalanine catabolism [42]. Similarly, previous studies have also indicated that structural genes in the flavonoid synthesis pathway may interact with each other to form an enzyme complex, participate in and regulate metabolic exertion, and ensure effective control of the metabolic flow [43]. For instance, the interaction between CHIL-CHS and CHS-CHI was confirmed in *Allium fistulous*, where CHIL forms a complex with CHS/CHI and interacts with membrane proteins anchored in the endoplasmic reticulum (such as F3′H) [44]. In Arabidopsis, the CHIL and CHI can interact directly with CHS [45], whereas and CHS and CHIL have been shown to interact in rice during flavonoid accumulation [46]. It has also been reported that the membrane-associated C4H cytochrome P450 may anchor the complex composed of PAL and other phenylpropane pathway enzymes to the endoplasmic reticulum (ER) [47,48,49]. In our study, CtC4H1-induced regulation of other key structural genes of flavonoid pathway demonstrated that the expression of the downstream regulatory genes including *AtF3H*, *AtDFR*, *AtFLS*, and *AtANS* showed an enhanced expression in OE lines than the WT. Interestingly, the expression of *AtPAL* was significantly increased in OE transgenic lines consistent with our previous findings of the protein interaction network between CtC4H and *CtPAL*. In addition, the expression of *AtDFR* and *AtANS* increased significantly, consistent with the concentration of anthocyanins in OE-CtC4H1 lines. Thus, these findings give important insights towards the interaction mode of CtC4H and CtPAL genes, which has led us to unleash the regulatory network of flavonoid biosynthetic pathway and antioxidant defense system in safflower.

## 4. Materials and Methods

### 4.1. Plant Materials, Treatments, and Growth Conditions

The seeds of safflower cultivar ‘Jihong NO.1′ were soaked for 5 h to remove the upper floating layer. After surface sterilization using HgCl2 solution (0.1%, *w*/*v*) for 10 min and washing 5–6 times with distilled water, the safflower seeds were germinated on wet filter paper at 25 °C for 72 h under sterile conditions. Then, those with uniform size were transplanted in plastic pots (20 cm × 25 cm), and eight plants per pot were prepared filled with black soil: vermiculite (7:3). The seedlings were cultured in an artificial climate chamber in a bioreactor at Jilin Agricultural University, and four seedlings were interspersed in each pot after one week of germination. The air temperature and humidity were set from 25 °C to 27 °C and 60% to 65% during the day and 18–21 °C and 60% to 65% during the night at 16/8 h light/dark conditions. For safflower seedlings that have grown for about 23 days, we set up four different stress treatments, including drought stress (20% PEG-6000, Solarbio, Beijing, China), methyl jasmonate (MeJA, Solarbio, Beijing, China) (300 μmol L^−1^), and light and dark. The petals of safflower were collected at different stages of flower development. All samples were placed in liquid nitrogen immediately after collection, and then stored at −80 °C until further use.

The Arabidopsis seeds were germinated in MS (Murashige and Skoog, Sigma, Saint Louis, USA) medium and then transplanted into a soil mix (vermiculite: good quality soil = 7:3 (V/V). We induced uniform drought stress by withholding water availability to Arabidopsis seedlings of WT, CtC4H1 overexpression (OE-4 and OE-10) lines and CtPAL1 overexpression (OE-6) lines when they reached the four-week-old stage. The drought stress treatment was continued for about two weeks until there significant changes appeared in phenotype, and then fresh leaves samples were collected from the experimental group and stored at −80 °C until next use. Growth conditions for Arabidopsis plants were maintained at a controlled temperature of 23 ± 2 °C, a 16/8 h light/dark photoperiod, and relative humidity of 70%.

### 4.2. RNA-Seq, Functional Annotation and Differential Expression Analysis

Total RNA was extracted from the petals of safflower at different flowering stages using RNAiso Plus reagent (Takara, Beijing, China) according to the instructions of the manufacturer, with slight modifications. Three biological replicates were sequenced using paired-end sequencing with a read length of 150 bp on an Illumina NovaSeq 6000 platform (Illumina, San Diego, CA, USA). To acquire high-quality reads, raw read quality was evaluated using FastQC with default settings, and adapter sequences were trimmed using CutAdapt. The safflower transcriptome was assembled with the Trinity software program. Cor-set was used to cluster assembled contigs based on common reads of transcripts, and typical sequences extracted from each cluster were designated as unigenes. To determine the functional annotation of unigenes, distinct sequences were mapped against many protein and nucleotide databases such as Nr, Nt, Pfam, COG, and Swiss-Prot. WeGo software was utilized for gene ontology functional categorization of the GO. The FPKM (Fragments Per Kilobase of Transcript Sequence per Million Base Pairs Sequenced) technique was used to calculate transcript expression levels. False Discovery Rate (FDR) control was employed to adjust the *p*-value in multiple tests conducted with DEseq2. We identified differentially expressed genes (DEGs) with|log2 (Fold Change)| > 1 and FDR 0.05.

### 4.3. Multiple Sequence Alignments and Phylogenetic Analyses

For multiple sequence alignment, the amino acid sequence encoded by CtC4H was used to screen homologous sequences from other species using NCBI blast search (https://BLAST.ncbi.nlm.nih.gov (accessed on 1 January 2021)) and TAIR website (http://www.arabidopsis.org/ (accessed on 1 January 2021)). The plant species selected for alignment of C4H amino acid sequences included *Chrysanthemum boreale*, *Camellia sinensis*, *Arabidopsis thaliana*, *Arctium lappa*, *Helianthus annuus*, *Nicotiana tabacum*, and *Ageratina adenophora* using MEGA 7.0 software. Then, a phylogenetic tree was constructed based on the neighbor connection method in MEGA software, and the bootstrap was set to test 1000 replicates. Protein interaction prediction was carried out using the amino acid sequence of CtC4H in STRING online network tool (https://string-db.org (accessed on 1 January 2021)).

### 4.4. Molecular Cloning and Subcellular Localization of CtC4H1

Total RNA was extracted from the petals of safflower following the aforesaid protocol. A NanoDrop 2000 (Thermo Fisher Scientific, Waltham, MA, USA) ultraviolet spectrophotometer and a gel electrophoresis were utilized to determine the concentration and quality of the extracted RNA. The cDNAs were reverse transcribed using a PrimeScript RT Reagent Kit (Takara, Beijing, China), following the Manufacturer’s instructions. The specific primers were designed (Appendix A) and synthesized based on the CtC4H1 sequence obtained from the safflower genome to derive the complete ORF of CtC4H1 gene. The PCR-amplified DNA clones were ligated onto T-easy cloning vectors and the positive clones were selected for sequencing. The CDS sequences of CtC4H1 and CtPAL1 genes were extracted and the cloning was performed using their specific pair of primers with added BamHI restriction sites. The plant overexpression vector (pCAMBIA1300-35S-GFP) was constructed for both genes. The recombinant plasmids of pCAMBIA1300-35S-CtC4H1-GFP and pCAMBIA1300-35S-CtPAL1-GFP were prepared and efficiently transformed into a competent Agrobacterium strain (EHA105) using the heat shock method. After confirmation with colony PCR, the bacterial strains were selected on YEP liquid medium containing the appropriate antibiotics and 100 μmol L^−1^ and incubated until the OD600 reached 0.8–1.0. The bacterial strains were collected using centrifugation at 5000 rpm for 15 min and then resuspended in a resuspension solution, which was prepared as follows (10 mM MgCl2 and 10 mM MES were prepared as follows: 1:1, autoclaved placed at 121 °C for 20 min, 100 μmol L^−1^ AS was added to the sterilized resuspension). The resuspension was incubated in the dark for 2–3 h and then injected from the abaxial surface of the tobacco leaves using a disposable medical syringe. The empty (pCAMBIA1300-35S-GFP) was used as a control vector for comparison. Tobacco leaves were taken after dark incubation of the injected tobacco for about 48–72 h, to observe and image green fluorescence under a laser confocal microscope following the instructions given by [46].

### 4.5. Yeast-2-Hybrid (Y2H) Assay and Bimolecular Fluorescence Complementation (BiFC)

The fragments of CtC4H1 and CtPAL1 were simultaneously inserted into GAL4 DNA binding domain vectors (pGBKT7, pGADT7) in order to generate recombinant pGBKT7-CtC4H1 and pGADT7-CtPAL1 plasmids. After confirmation using restriction digestion and sequencing, the recombinant plasmids were successfully transformed into Y2HGold yeast receptor cells according to the manufacturer’s protocol. The transformed cultures were cultured separately in SD/-Leu-Trp medium and incubated at 30 °C for 3–5 days before transferring the yeast colonies to a SD/-Ade-His-Leu-Trp medium. The growth of each strain was observed after 3–5 d. LacZ reporter genes were additionally detected using the X-α-gal-containing color development solution [50]. Similarly, pxy106-CtC4H1 and pxy104-CtPAL1 recombinant plasmids were transferred to the EHA105 Agrobacterium receptor cells, and single colonies were subjected to growth on the YEP solid medium containing Spec+ antibiotic. The positive colonies were picked and tested using the PCR system. The bacteria were sub-cultured on YEP liquid medium containing Spec+, Rif+, and 100 μmol L^−1^ at an OD600 of about 0.8–1.0. and was extracted with centrifugation at 5000 rpm for 15 min. The 5-week-old tobacco leaves were incubated with the recombinant strains in the dark for 48–72 h, and then the yellow fluorescence imaging was observed under a laser confocal microscope [51].

### 4.6. Construction of the Plant Expression Vector and Arabidopsis Transformation

The pBASTA overexpression vector containing a selectable marker gene (BAR) under the control of the 35S (CaMV) promoter was constructed by ligating a full-length fragment of BamHI/EcoRV-digested CtC4H1 into the expression cassette of the pBASTA vector using T4 ligase according to the instructions. The recombinant plasmid (pBASTA-CtC4H1) produced after ligation was confirmed primarily with double restriction digestion and then transformed into *E. coli* DH5α cells using a conventional heat shock protocol. Sequence integrity of PCR-positive bacterial cells from the colony was then confirmed with Sanger sequencing. A recombinant plasmid (pBASTA-CtC4H1) construct was generated under the control of the 35S viral promoter, and a separate empty plasmid of pBASTA was inserted into *A. tumefaciens* (EHA105) receptor cells. Positive colonies of EHA105 were selected on the YEP+ antibiotic (Sodium carbamate, rifampicin) growth medium and further confirmed with colony PCR methods. Floral dip transformation of WT *A. thaliana* plants was carried out according to the instruction given by [45]. For the qRT-PCR assay, two different types of experimental lines were selected, in which the WT and two CtC4H1 independent transgenic OE lines of *A. thaliana* were selected. The total RNA content of the WT, CtC4H1-OE lines were extracted simultaneously using a previously described protocol. The cDNA templates were prepared using the PrimeScript RT Reverse Transcription Kit (Takara, Beijing, China), following the protocol given by the manufacturer.

### 4.7. Expression Analysis

The total RNA content was extracted from safflower and Arabidopsis under specific conditions were extracted following the previously described method. The reverse transcriptase superscript IV kit (Thermo Fisher Scientific, Waltham, MA, USA) was used to prepare the first strand cDNA template from the RNA of each experimental material and the cDNA was diluted up to 100 times before qRT-PCR reaction. The forward and reverse pair of primers for each assay was prepared using premier 3 software and the details are listed in Appendix A. Then, qRT-PCR assay was carried out using SYBR Green-Mix (Bio-Rad, Hercules, CA, USA) kit. The expression level was normalized using the expression level of 18s ribosomal RNA gene (AT5G38720.1). The relative fold change values were calculated using the method of 2^−ΔΔCt^.

### 4.8. Determination of Total Flavonoids and Anthocyanins Content

The total flavonoids content in transgenic plants were quantified using the previously described method [3] with minor modifications. An accurate weight of 0.1 g leaves of Arabidopsis thaliana was ground in liquid nitrogen until in powder form. The powder sample was transferred to a 10 mL centrifuge tube, and 0.5 mL of methanol (Solarbio, Beijing, China) added. Then, 0.15 mL of 5% sodium nitrite solution (Solarbio, Beijing, China) was added and left for 6 min followed by the addition of 2 mL of 4% sodium hydroxide solution (Solarbio, Beijing, China). After that, 0.15 mL of 10% aluminum nitrate solution (Solarbio, Beijing, China) was added and left for 6 min. Then, distilled water was added to fix the volume to 5 mL, shaken and mixed, and left for 3 min. Finally, the sample was centrifuged at 10,000× *g* for 10 min, and 0.3 mL of supernatant was transferred to a spectrophotometric plate and the sample was analyzed at 508 nm. The TFC content was calculated according to the regression curve drawn for the rutin standard. Similarly, the anthocyanins content was measured according to the method described by [4,38] with slight modifications. An equal weight of 0.1 g leaves of Arabidopsis was ground quickly to a powder in liquid nitrogen, and then 2 mL of extract (97:3 volume ratio of methanol to hydrochloric acid) was added. After shaking and mixing the samples, the incubation cycle was conducted in the dark at 4 °C for 16 h in the refrigerator. Then, the samples were centrifuged at 12,000 rpm for 10 min at 4 °C, and the supernatant was collected. Finally, the absorbance of each sample was measured at 530 and 657 nm. The total anthocyanin content (µg·g^−1^) of each sample was calculated according to the following formula: (A530 − 0.25 × A657) × 449.2 × 2 × 100/(26,900 × 0.1).

### 4.9. DAB and NBT Staining

To investigate hydrogen peroxide (H_2_O_2_) and superoxide anion (O_2_^−^) content in OE-CtC4H1 lines and WT plants after drought treatment, 3,30-diaminobenzidine (DAB, Solarbio, Beijing, China) tetrahydrochloride hydrate and nitroblue tetrazolium (NBT, Solarbio, Beijing, China) staining were used, respectively. Plant material was placed in 1 mg mL^−1^ DAB staining solution (pH 5.7) and 0.5 mg mL^−1^ NBT staining solution (pH 7.5) and then stained overnight (approximately 12 h) at room temperature, under the protection from light during staining. The stained leaves were transferred to 95% ethanol and heated in a boiling water bath for 15 min to discolor the leaves. After cooling, the leaves were replaced with fresh anhydrous ethanol and stored in a refrigerator at 4 °C. The results were observed with inverted microscopy.

### 4.10. Measurement of Physiological Traits

The malondialdehyde (MDA) content was measured using the thiobarbituric acid-reactive substances (TBARS) assay with slight modifications. A total of 0.1 g of plant sample was subject to a solution containing 1 mL of 10% (*w*/*v*) trichloroacetic acid (TCA, aladdin, Shanghai, China) and placed in an ice bath until complete homogenization. The resultant solution was transferred to a 5 mL tube and then centrifuged at 12,000× *g* for 15 min at 4 °C. Then, 1 mL of supernatant was placed into another clean 5 mL centrifuge tube, and 1 mL of 0.6% thiobarbituric acid (TBA, aladdin, Shanghai, China) solution was immediately added. After mixing, the boiling step was carried out for 20 min and then immediately cooled on ice and finally centrifuged at 12,000× *g* for 15 min at 4 °C. The absorbance values at 450 nm, 532 nm, and 600 nm were measured for MDA content. The concentration of MDA per unit fresh weight was then calculated according to the formula C/μmol/L = 6.45(A532 − A600) − 0.56A450. Similarly, the superoxide dismutase (SOD), catalase (CAT) and peroxidase (POD) activities were measured via the test kits (Solarbio, Beijing, China). The content of proline, hydrogen peroxide (H_2_O_2_) and superoxide radical (O_2_^−^) were measured following the methods describe by [52,53].

### 4.11. Statistical Analysis

Statistical analysis data were presented as mean ± standard deviation of three independent biological replicates. All statistical analyses were performed using GraphPad prism v.9.1.3. Firstly, the normality and variance of each data set were tested and determined, and the data were analyzed through the analysis of variance (ANOVA) and other parameter tests. For analysis of variance, if the test statistics were significant, Tukey’s multiple comparison post-test was performed. Asterisks indicate statistical significance (*, *p* < 0.05; **, *p* ≤ 0.01; ***, *p* ≤ 0.001). Graphics were created and organized using the GraphPad prism.

## 5. Conclusions

In this study, the identification and overexpression of the CtC4H1 gene in Arabidopsis has led us to understand its positive role in drought stress tolerance by regulating flavonoid accumulation and the antioxidant defense system. The enhanced accumulation of specialized metabolites in transgenic Arabidopsis enabled oxidative damage control of ROS, resulting in alleviated oxidative stress caused by drought stress. These findings lay the foundation work for future research focused on how secondary metabolic flux improves drought stress strategies in safflower.

## Figures and Tables

**Figure 1 ijms-24-05393-f001:**
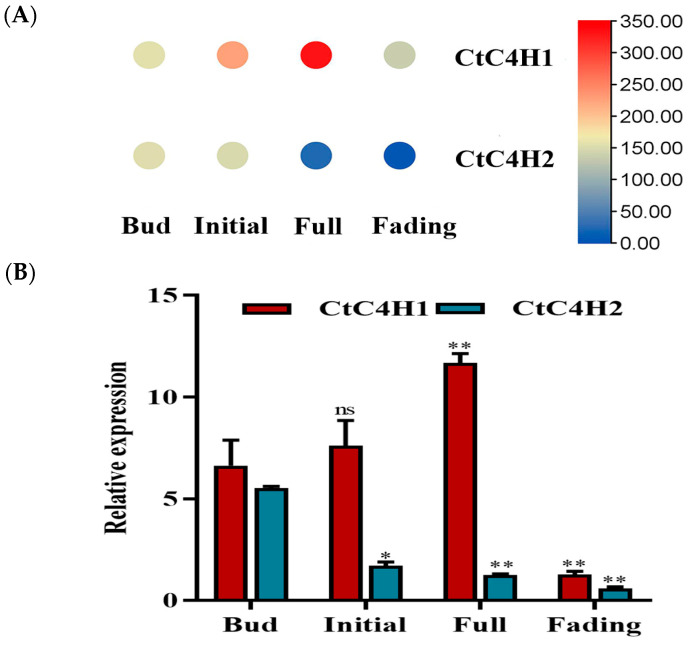
Expression profiling of two candidate CtC4H-encoding genes in different flowering stages of safflower. (**A**) The heat map demonstrates the expression level of CtC4H1 and CtC4H2 genes in four different flowering stages of safflower (bud, initial, full, and fading) using the FPKM ratios obtained from transcriptomic data. (**B**) The validation of the relative expression level of CtC4H1 and CtC4H2 genes in different flowering stages of safflower was determined using qRT-PCR analysis. The statistics were calculated with one-way ANOVA test and significant differences between treatments were demonstrated with asterisk (*ns* no significant, ** p* < 0.05, ** *p* < 0.01).

**Figure 2 ijms-24-05393-f002:**
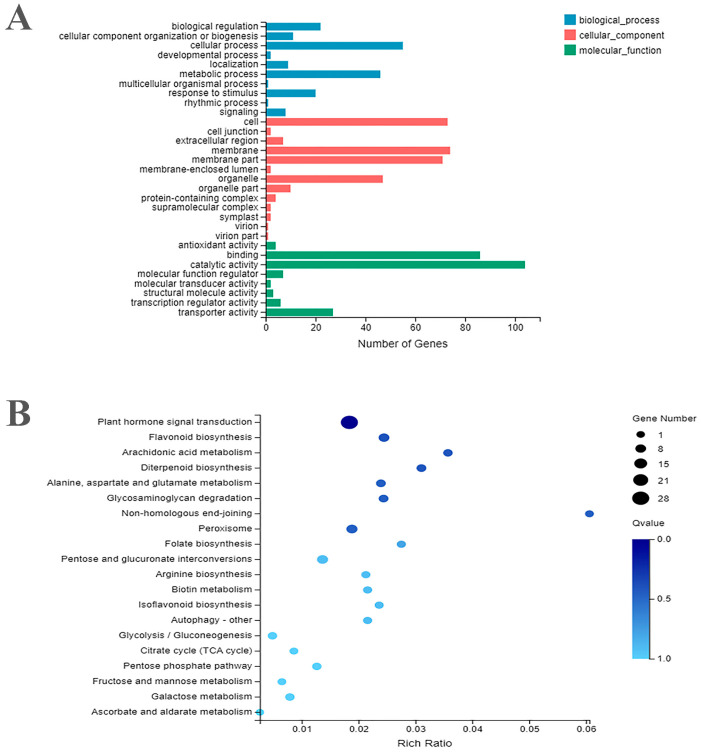
Functional annotations and KEGG pathway enrichment analysis of differentially expressed genes identified from safflower. (**A**) GO terms classification into three main categories: MF, CC, and BP. The distribution of genes in each term is shown alongside the *x*-axis (**B**) The scatterplot of the top 20 enriched KEGG pathways in safflower. The *x*-axis shows the rich ratio of DEGs to the total number genes in each pathway.

**Figure 3 ijms-24-05393-f003:**
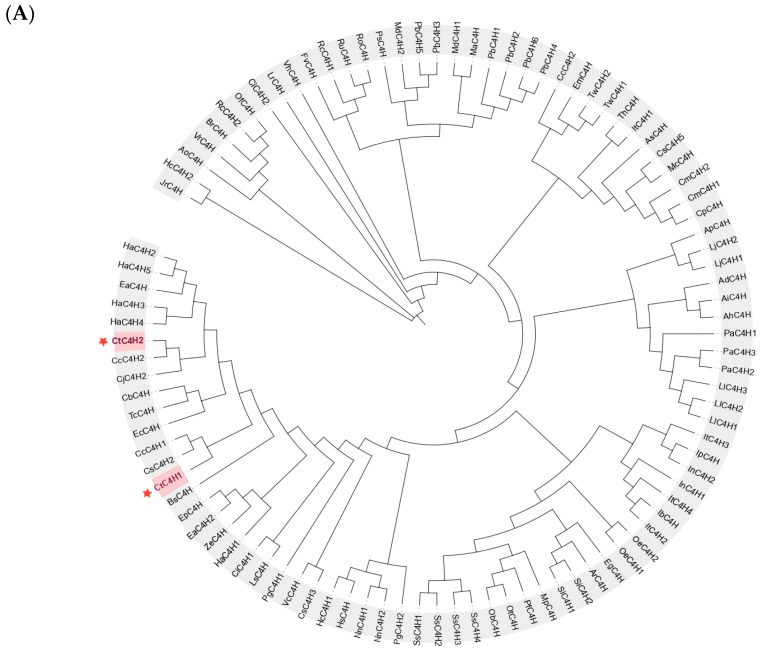
Phylogenetic analysis and multiple sequence alignment of C4Hs proteins (**A**) The phylogenetic tree containing C4H members from safflower and other plant species. The tree was constructed with MEGA 6.0 using the neighbor-joining (NJ) method and 1000 bootstrap replications. (**B**) Multiple sequence alignment of CtC4H1 and C4H from other plant species. The P450 conserved sequences are boxed in red, the underlined regions represent the six SRSs and Triangle represents E-R-R triad; Cb: *Chrysanthemum boreale*; Cs: *Camellia sinensis*; At: *Arabidopsis thaliana*; Al: *Arctium lappa*; Ha: *Helianthus annuus*; Nt: *Nicotiana tabacum*; Aa: *Ageratina adenophora*.

**Figure 4 ijms-24-05393-f004:**
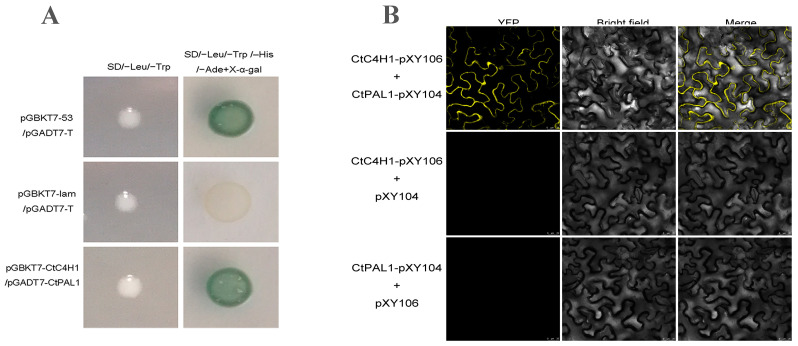
The interaction of CtC4H1 and CtPAL1 in tobacco (**A**) Yeast-two-hybrid assay of CtC4H1 and CtPAL1. (**B**) BiFC analysis. Fluorescence was observed in tobacco leaf epidermal cells. Scale bars = 25 μm.

**Figure 5 ijms-24-05393-f005:**
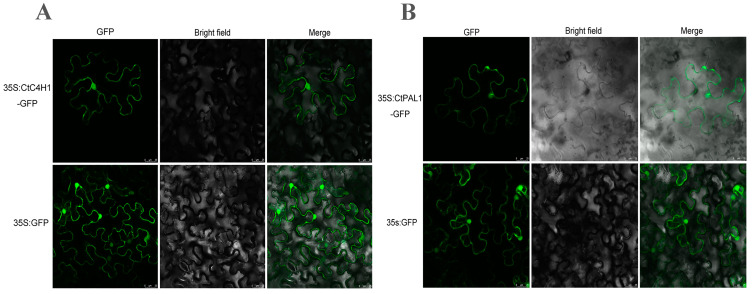
Subcellular location of CtC4H1 and CtPAL1 using GFP fusion constructs in tobacco leaves. (**A**) The green, fluorescent signals of the pCAMBIA1300-CtC4H1-GFP and (**B**) pCAMBIA1300-CtPAL1-GFP. The vector construct containing GFP alone was used as a control. Bars, 25 µm.

**Figure 6 ijms-24-05393-f006:**
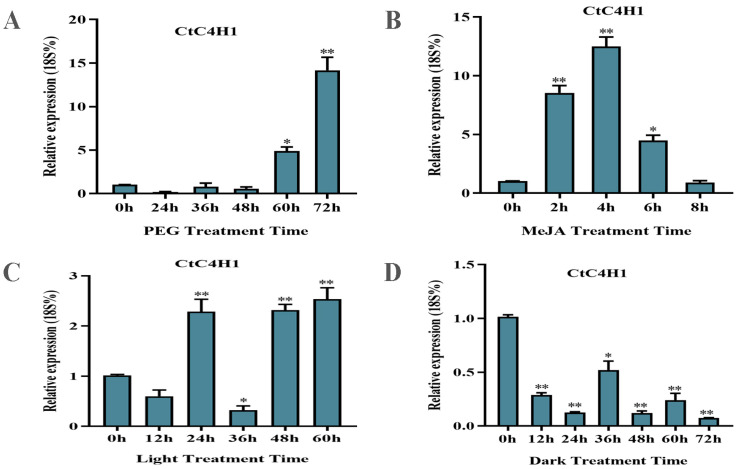
Expression analysis of CtC4H1 gene under different abiotic stress conditions. (**A**) PEG treatment. (**B**) MeJA treatment. (**C**) Light treatment and (**D**) dark treatment. The relative expression level was normalized with 18S rRNA gene which was used as housekeeping gene. The numerical value represents the average ± standard deviation of three biological repetitive samples. The statistics were calculated with one-way ANOVA test and significant differences between treatments were demonstrated with asterisk (* *p* < 0.05, ** *p* < 0.01).

**Figure 7 ijms-24-05393-f007:**
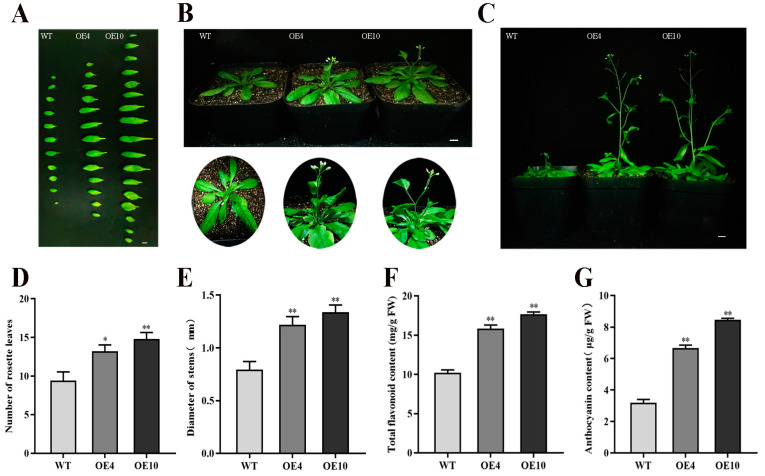
Phenotypic analysis and quantification of total flavonoids and anthocyanin content in two transgenic OE lines of Arabidopsis (OE-4 and OE-10) and WT. (**A**) Leaf morphology of CtC4H1 transgenic OE and WT plants (**B**) phenotypic analysis of CtC4H1 transgenic OE and WT seedlings in the soil (**C**) stem growth of CtC4H1 transgenic OE and WT plants. Bars = 1 cm (**D**) Number of rosette leaves. (**E**) Diameter of stems. (**F**) Total flavonoids content of WT, CtC4H1 transgenic lines. (**G**) Anthocyanin content of WT, CtC4H1 transgenic lines. The numerical value represents the average ± standard deviation of three biological repetitive samples. The statistics were calculated with one-way ANOVA test and significant differences between treatments were demonstrated with asterisk (* *p* < 0.05, ** *p* < 0.01).

**Figure 8 ijms-24-05393-f008:**
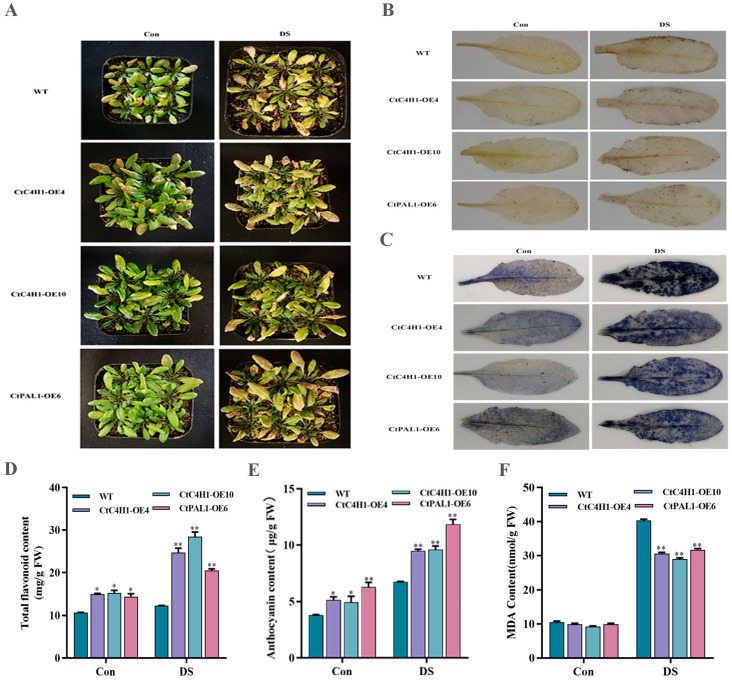
Leaves phenotypic and biochemical analysis of CtPAL1 and CtC4H1 transgenic plants after DS (drought stress) treatment. (**A**) Phenotype of transgenic plants after 14 days of drought stress treatment. (**B**) DAB staining assay, leaves were stained with DAB after drought treatment. (**C**) NBT staining assay, leaves were stained with DAB after drought treatment. (**D**) Total flavonoid content. (**E**) Anthocyanin content. (**F**) Malondialdehyde. (MDA) content analysis. The numerical value represents the average ± standard deviation of three biological repetitive samples. The statistics were calculated with one-way ANOVA test and significant differences between treatments were demonstrated with asterisk (* *p* < 0.05, ** *p* < 0.01).

**Figure 9 ijms-24-05393-f009:**
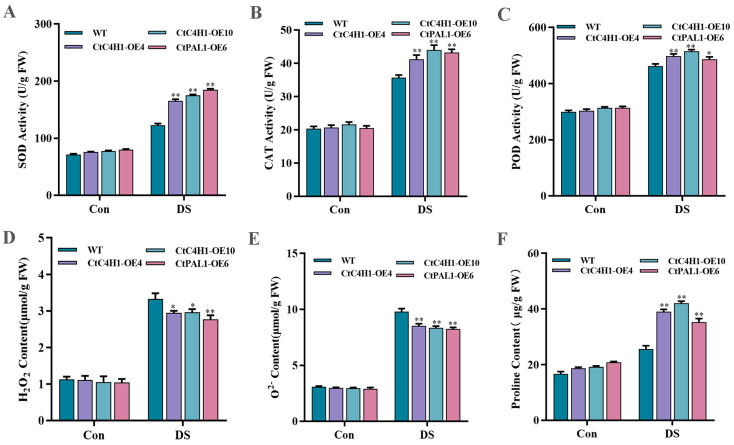
Determination of antioxidant enzymes activities and related physiological indexes of CtPAL1 and CtC4H1 transgenic plants after drought stress treatment. (**A**) Superoxide dismutase activity (SOD). (**B**) Catalase activity (CAT). (**C**) Peroxidase activity (POD). (**D**) H_2_O_2_ content analysis. (**E**) O_2_^−^ content analysis. (**F**) Proline content analysis. The numerical value represents the average ± standard deviation of three biological repetitive samples. The statistics were calculated with one-way ANOVA test and significant differences between treatments were demonstrated with asterisk (* *p* < 0.05, ** *p* < 0.01).

**Figure 10 ijms-24-05393-f010:**
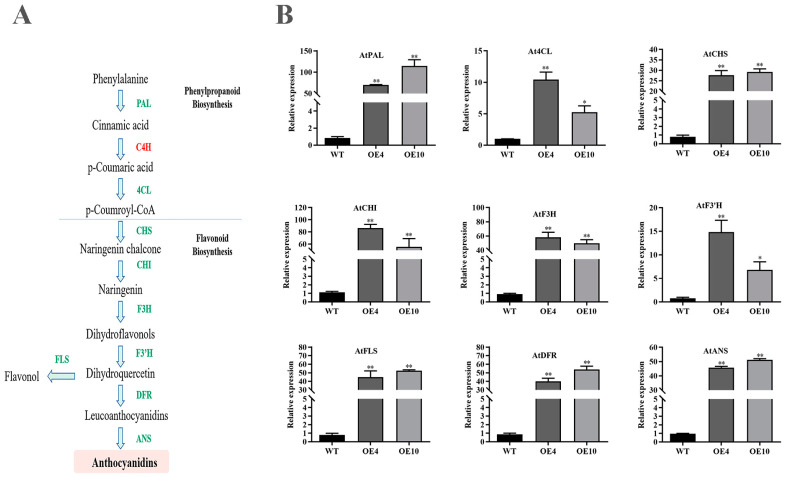
The expression of genes related to flavonoid synthesis pathway in CtC4H OE transgenic lines. (**A**) Schematic diagram of flavonoid synthesis pathway. (**B**) The qRT-PCR analysis of key genes involved in flavonoid biosynthesis (AtPAL, At4CL, AtCHS, AtCHI, AtF3H, AtF3′H, AtFLS, AtDFR, AtANS) in WT and CtC4H1 transgenic plants. The numerical value represents the average ± standard deviation of three biological repetitive samples. The statistics were calculated with one-way ANOVA test and significant differences between treatments were demonstrated with asterisk (* *p* < 0.05, ** *p* < 0.01).

## Data Availability

The sequencing data has been deposited into NCBI under the accession number PRJNA399628. All data generated or analyzed during this study are included in the published article and Appendix A.

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
