# Peer review of "A Cinnamate 4-HYDROXYLASE1 from Safflower Promotes Flavonoids Accumulation and Stimulates Antioxidant Defense System in Arabidopsis"

_ijms, 2023, doi:10.3390/ijms24065393_

Round 1
Reviewer 1 Report
In this manuscript, the authors investigated the expresison of a safflower C4H1 at different stages and conditions, and its possible interaction with CtPAL1, which is another key player in flavonoid biosythesis. Using a transgenic Arabidopsis system, the authors characterize the function of CtC4H1 and CtPAL1 in drought stress responses and the protection against oxidative stress. This comprehensive study would broaden the understanding of drought stress-responsive menchanism in safflower. Several concerns need to be addressed before publication.
1. In the subcellular localization experiment, a plasma membrane marker construct is needed for co-expression to support that CtC4H1 and CtPAL1 are localized at the plasma membrane.
2. To support the effect of overexpressing CtC4H1 and CtPAL1 in Arabidopsis, other parameters such as RWC should be measured. The current figure does not display a clear difference between the OE plants and wild type.
3. The degree of GO enrichment is not shown in the figure. The authors should also describe the threshold for the GO enrichment analysis.
4. In the KEGG analysis, the terms with q-value>0.05 have little scientific meaning.
5. The figures are distorted or not fully displayed.
6. It is interesting that overexpressing CtC4H in Arabidopsis led to high expression of the genes involved in the flavonoid synthesis pathway. The author should discuss more and provide some possible explaination with reference to literature. In addition, it is worth analyzing the transcriptome data generated to investigate the co-expression pattern of all key genes in the synthesis pathway in Safflower.
7. In Line 736, the two reference do not include methods for measuring the content of proline, hydrogen peroxide and superoxide radical. Please check the reference carefully.
8. The transcriptomic data should be deposited to a public database.
Reviewer 2 Report
The authors examine an aspect of drought tolerance little studied the role of the phenolics
thus the work has originality - of value is the fact that they do also show results for the more standard approach proline, ROS and ROS deg enzymes
Several comments made - format requires attn in places but nothing serious
Would have liked to have seen more controls such as results with empty expression vectors in the transgene studies
more comments too on how the normal light regime would affect data sets ie was there care always to harvest material at a certain time of light exposure

Round 2
Reviewer 1 Report
The authors have made efforts to improve the manuscript.
As drought tolerance is a main focus of this study, it is better to provide quantitative parameter to support the enhanced drought tolerance, which is not clearly shown by the plant photos. Otherwise, it is safer to just emphasize that CtC4H1 stimulates antioxidant defense system under drought stress.
Author Response
Dear editor and reviewer,
Thank you very much for providing us with minor revisions on our revised manuscript. We agree with the reviewer’s suggestion that a clear phenotypic changes are still not evident to support drought stress tolerance. However, we have alternatively provided several quantitative analysis apart from the antioxidant enzymatic system which strongly support our hypothesis. For example the analysis of qRT-PCR analysis for expression analysis under drought stress, DAB staining of transgenic leaves under drought stress, metabolite accumulation as well as the expression induction of other pathway genes, predominantly supporting our research hypothesis of CtC4H-induced drought stress tolerance. The prevailing arguments were then supported by additional experiments related to antioxidant enzymatic system and therefore we came to the conclusions that CtC4H facilitates drought stress tolerance in Arabidopsis and safflower.